# Effect of High-Oleic Peanut Intake on Aging and Its Hippocampal Markers in Senescence-Accelerated Mice (SAMP8)

**DOI:** 10.3390/nu12113461

**Published:** 2020-11-11

**Authors:** Kiharu Igarashi, Daisuke Kurata

**Affiliations:** 1Department of Bioresource Engineering, Faculty of Agriculture, Yamagata University, 1-23 Wakaba-Machi, Tsuruoka, Yamagata 997-8555, Japan; 2Bean Research Section, Denroku Co Ltd., 3-2-45, Kiyozumi-Machi, Yamagata 990-8506, Japan; daisuke-kurata@denroku.jp

**Keywords:** high-oleic peanut, aging, cognitive impairment, hippocampus, GABA/Glu ratio, proteomics, SAMP8

## Abstract

In many previous studies, the preventive effects of peanut against aging and cognitive impairment have often been unclear, so to clarify the effects we first investigated effective markers for evaluating its effects in the hippocampus of senescence-accelerated mouse prone/8 (SAMP8) mice, mainly using proteomics. The effects of dietary high-oleic peanuts on the hair appearance of SAMP8, the expression of effective markers in the hippocampus, and the TBARS and amino acid contents of the hippocampus were examined. Hippocampus solute carrier family 1 (glial high-affinity glutamate transporter), calcium/calmodulin-dependent protein kinase type II, and sodium- and chloride-dependent GABA transporter, which all are considered to be closely related to glutamic acid concentration were decreased by feeding of the samples, and the GABA/glutamic acid ratio in the hippocampus was increased by feeding with the samples. The formation of glial fibrillary acidic protein and synapsin-2, which showed higher levels in the SAMP8 than in SAMR1, and the protein expression of tyrosine 3-monooxygenase/tryptophan 5-monooxygenase activation protein and dihydropteridine reductase, which are considered to be related to the formation of adrenergic neuron transmitters, were reduced by the feeding of peanuts and their germ-rich fraction. Ferulic acid, as an ester and minor component in peanuts, could be partly connected to the effect of peanuts. These results indicate that high-oleic peanuts and their germ-rich fraction can protect against aging and cognitive impairment by regulating protein expression, which could be measured by the proteomics of the above hippocampus proteins of SAMP8 and the hippocampal GABA/glutamic acid ratio.

## 1. Introduction

Roasted peanuts, which are widely consumed as cakes containing sugars and/or other food additives, are favored by many people in Japan because of their characteristic taste and flavor, and the number of products made with them is increasing. However, since most of the germ is detached from wholegrains in the process of dehulling the astringent inner skin of the roasted peanuts by grinding, only parts of cotyledon and the other parts without germ are used for cake making. The development of methods for the utilization of the germ as a food material for other food production, as well as in functional foods, is being investigated in the food industry.

It has been pointed out that intake of peanuts with high-oleic acid may improve cerebrovascular and cognitive function [1], and it has also been pointed out that oligomeric proanthocyanidins, which are contained in peanut skins and have anti-oxidative activity, may improve memory deficits in aging people and extend the lifespan of senescence-accelerated mouse prone/8 (SAMP8) [2]. Further, it is reported that walnuts containing a number of potential neuroprotective compounds, such as vitamin E, folate, flavonoids, and ellagic acid, are effective in improving memory deficits in transgenic mouse models of Alzheimer’s disease. These reports suggest that peanuts, which may also contain the same compounds as walnuts, may be able to protect against Alzheimer’s disease, which accompanies aging [3]. The improvement in cognitive function in older women who consume nuts at a higher volume for a longer time may support the benefit of nuts in protecting against cognitive function decline and aging [4,5]. This information prompted us to study whether high-oleic peanuts are also available for protection against aging and cognitive impairments, and whether the germ-rich fraction of high-oleic peanuts released as a by-product during processing also has the same effect as the wholegrains of roasted peanuts, including their action mechanism in SAMP8.

For the first step in this study, markers which are able to evaluate the effect of samples on reducing aging and cognitive impairment were selected by an analysis of amino acids and a proteome analysis of the hippocampus of SAMP8 and senescence-accelerated resistant mice (SAM-R1). Then, the effects of the samples on the markers were evaluated and compared among the samples. There have been some reports that clearly showed the relation between glutamic acid and other amino acid contents in the hippocampus and cognitive disease or aging [6,7]. We then determined the effects of the samples in a proteomic analysis of the hippocampus. There have been many reports evaluating the effect of food components using a few markers [8,9,10,11,12,13]. Superoxide dismutase (SOD), tubulin alpha, hemoglobin subunit alpha, glutathione S-transferase, synapsin-2, and dihydropteridine reductase were tentatively settled on as hippocampus markers, because it is reported that the amounts of these enzymes or proteins change in the rat hippocampus with aging [6]. The protective effects of ferulic acid, which is contained in peanuts in the form of esters, were also examined as a reference and to investigate whether the components could contribute to the effect of high-oleic peanuts.

## 2. Materials and Methods

### 2.1. Sample Preparation

Peanuts (Runner species) with a high-oleic acid content (oleic acid content of over 70%) were obtained from Texas, America, and roasted for several minutes. The part detached from the whole part of peanuts during the dehulling of the astringent inner skin of the roasted peanuts by grinding was used as the germ-rich fraction in this experiment. The lipid contents of the whole peanuts and the germ parts were 50.2% and 44.0%, respectively. The carbohydrate, protein, lipid, ash, and moisture percentages of the roasted whole peanuts were 19.6%, 25.5%, 50.2%, 2.3%, and 2.2%, respectively, and those of the germ parts were 26.0%, 23.6%, 44.0%, 2.8%, and 3.2%, respectively. Ferulic acid detected as a minor component of peanuts after its alkaline hydrolysis with 2N-NaOH at 30 °C for 30 min was also used in this experiment. It was purchased from Sigma-Aldrich Japan (St. Louis, MO, USA) and used for the animal experiment.

### 2.2. Animals and Diet

Six-month-old senescence-accelerated mice (SAMP8) and senescence-accelerated resistance mice (SAMR1) as a control were purchased from the Shizuoka Laboratory Animal Center (Hamamatsu, Japan) and cared for according to the institutional guidelines of Yamagata University. The approval no. of the Institutional Animal Care and Use Committee of Yamagata University was 2905 (IACUC number: 2905). Mice were housed individually under a 12:12 h light–dark cycle at 22 ± 2 °C and 40–60 humidity. After acclimating for 3 days, the mice were randomly divided into six groups of five or six mice and fed either a basal diet (SAMR1 = R1 and SAMP8 = P8 groups, *n* = 6), basal diet with peanuts (wholegrain) (SAMR1 + H = R1 + H and SAMP8 + H = P8 + H group, *n* = 6), basal diet with germ from peanuts (SAMP8 + G = P8 + G group, *n* = 6), or basal diet with ferulic acid (SAMP8 + F = P8 + F group, *n* = 5). The composition of each experimental diet is shown in Table 1. The oil level in each diet was reduced to 7% by decreasing the corn oil added to the R1 + H, P8 + H, and P8 + G groups. The diet and water were given ad libitum. Body weight and food intake were measured every 3 days. Learning and memory ability tests were performed with the step-through test system (Shock generator: MK-SG05, Muromachi Kikai Co., Ltd., Tokyo, Japan) on day 46–47 during feeding. For the first time, mice were put in a light box, which was partitioned by an opener from a black box, then the time required to move from the light box to the black box was measured (the first time). The mice moved to the dark box were soon exposed to electric stimulation at 0.08 mA for 0.3 s per mouse. After the time required to move to the black box was measured, the mice were put into the light box again, the same as the first measurement time. This procedure was repeated 3 times. The duration of time for movement to the black box was considered to be a learning ability. After the three learning tests, the mice were put into the breeding cages without electric stimulation, then after one and two days the mice were again put into the light box. The time required to move to the black box was measured as a memory ability. The learning test and memory availability test were set to finish one week before the dissection date to avoid stress to the mouse as much as possible and so as not to affect various measurement data after the dissection.

The method used for evaluating the degree of senescence in mice was the same as the method of Hosokawa et al. [14]. Hosokawa et al. used eleven categories associated with age-associated changes in behavior and appearance and evaluated their degree of severity in grades, but in this experiment only differences in appearance, such as glossiness, coarseness, loss of hair, skin ulcers, and lordokyphosis of the spine (each ranked in five grades, from 1 to 5, depending on the severity of symptoms), were evaluated as a score, and the sum of the grade was calculated as an aging score.

At the end of the feeding period, the mice were anesthetized with isoflurane 3–5 h after depriving their diet. After the collection of blood from the heart with a syringe, the brain was detached. The brain hippocampus was separated from the detached brain as soon as possible. The hippocampus was stored at −80 °C until analysis.

### 2.3. Amino Acid, TBARS, and Proteome Analyses

The frozen hippocampus was homogenized in cold 80% methanol (MeOH) by grinding with a plastic pestle, then ultra-sonicated for 15 min in an ice bath, followed by centrifugation at 1.1000 rpm for 15 min to obtain a protein pellet and supernatant [15]. The pellet was re-homogenized with cold 80% MeOH and then centrifuged. This procedure was repeated three times. The protein pellet and the supernatant were used for proteomics and amino acids analyses, respectively. The supernatant was also used for the measurement of TBARS (Thiobarbituric acid reactive substances) by the method of Uchiyama and Mihara [16], in which the TBARS values were measured using the fluorometric method at Ex 532 and Em 560 nm after the formed malon dialdehyde (MDA)-TBARS complex was purified by moving it to an n-butyl alcohol layer. The amino acids contents were measured by the ninhydrin method using an amino acid analyzer (Hitachi L8800, Tokyo, Japan).

The protein pellet for proteomics was suspended using the RapiGest SF (Waters, Milford, MA, USA) solution, followed by the reduction and alkylation of the cysteine residue in protein (Zhao et al., 2010) [17] and then digestion with sequencing-grade modified trypsin (Promega, Madison, WI, USA) for 16 h at 37 °C. The digested sample solution was added to 10% TFA (0.5% TFA at the final concentration), followed by incubation at 37 °C for 30 min and centrifugation at 10,000 × *g* for 20 min at 5 °C. The obtained supernatant was freeze-dried and re-dissolved with a small amount of H_2_O, followed by centrifugation at 10,000 × *g* for 20 min at 5 °C. Finally, it was analyzed by a UPLC ESI-Q-TOF MS/MS system (Xevo Q-Tof MS, Waters, Manchester, UK).

The separation of peptides in the supernatant by the UPLC (nano AQUITY, Waters, Manchester, UK) was performed on a BEH C-18 column (2.1 × 100 mm, 1.7 µm; Waters, Ireland, UK) using a solvent system composed of 0.1% HCOOH in H_2_O (solvent A) and 0.1% HCOOH in acetonitrile (solvent B). A linear gradient of B in solvent A (100% of B at 45 min) was used. The spectra were acquired in positive ion mode in micro Q-TOF with a mass range of 50–2000. The source temperature was set as 120 °C. Protein identification was carried out using ProteinLynx (Waters, Milford, MA, USA), and a database of the mice (taxonomy ID: 10088) was provided by NCBI (National Center for Biotechnology Information, Bethesda, MD, USA). The carbamidomethylation of cysteine and oxidation of methionine were considered to be variable modifications of tryptic peptides in MS/MS analysis. Differential protein analyses were performed after data normalization using the ProteinLynxTM Gloval Server (version 2.3) (Waters, Manchester, UK).

### 2.4. Statistics

Each value is given as a mean ± SEM for six animals, except in the case of the SAMP8 + F group in which five animals were used. The homogeneity of variance between treatments was verified by Bartlett’s test. Data were statistically analyzed using a one-way analysis of variance. A post-hoc analysis of significance was performed by Tukey’s multiple range test. A comparison between two groups was carried out using the Student’s *t*-test. All comparisons were considered statistically significant at *p* < 0.05.

## 3. Results and Discussion

### 3.1. Suppression of Peanut on Aging of SAMP8 in Appearance (Phenotype) and Learning and Memory Abilities

Although the food intake of each group was not different statistically, the body weight of the R1 + H group showed a tendency to be higher than the other group from the middle to the latter half of the period (Figure 1).

It is reported that some of the food and its components could reduce brain amyloid beta accumulation, and cognitive deficits and promote the antioxidative defense system in SAMP8 [18]. However, the relation of the aging of SAMP8 with the appearance (phenotype), such as the coarseness and yellowing of the back coat and loss of hair, and decreased learning and memory abilities has never been precisely and systematically determined. The degree of senescence of mice (tentatively expressed as the aging score) given an experimental diet (Figure 2), which was calculated from the appearance of mice, such as the coarseness and yellowing of the back coat, loss of hair, and other characteristics [14], was higher in SAMP8 (see P8 group) and was the lowest in the SAMR1 group (see R1), and the yellowing of the back coat in the SAMP8 group (see P8) was suppressed most strongly by feeding them the germ-rich fraction (see P8 + G), followed by ferulic acid (see P8 + F) and peanut (see P8 + H) in a decreasing order. However, there were no differences statistically in the learning ability among the peanut- (wholegrain), germ-rich fraction-, and ferulic acid-fed mice (each group: P8 + H, P8 + G, P8 + F), in spite of the fact that the R1 mice and peanut-fed R1 mice (each group: R1, R1 + H) exhibited a higher learning ability than the SAMP8 (Figure 3). The reasons for these results may be the weak effects of the sample and a lowering in the responsibility of SAMP8, compared to SAMR1, for electric stimulation by aging. Experiments using stronger electric stimulation might be necessary for the confirmation of the effect of high-oleic peanuts.

### 3.2. Relation between (GABA/Glutamic Acid) Ratio as a New Senescense Marker and Peroxidation in the Hippocampus

Although glutamic acid is thought to play an important role in the functions of learning and memory, its abundant accumulation in the brain is believed to be involved in the pathogenesis of a variety of neurodegenerative disorders in which cognition is impaired [19]. This information suggests that the glutamic acid concentration in the brain may be a marker showing cognitive impairment and aging which could be used as a tool for the research and development of foodstuffs and their components available for the suppression of cognitive impairment and aging. On the other hand, it is well known that brain TBARS exhibits a higher value in SAMP8 than SAMR1 [20,21], where the former age more rapidly than the latter. Considering this information, relations between the concentration of hippocampus amino acids, the ratios of concentrations of some amino acids, and the hippocampus TBARS values were determined. The hippocampus TBARS values and the concentrations of Glu and GABA, in addition to the GABA/Glu ratio, are shown in Figure 4 and Figure 5, respectively. Although the TBARS value of the SAMP8 (P8) group did not differ from that of the SAMR1 (R1) group at a statistically significant level, it showed significantly lower levels in the R1 + H and P8 + H groups with added peanut wholegrains than the R1 and P8 groups, respectively. This result suggested that the feeding of peanuts (wholegrains) may be effective in protecting against the oxidation of the hippocampus. The amounts of glutamic acid in the hippocampus, which are reported and/or considered to be increased in the brain of cognitive mice [19], did not show any change with the feeding samples, which are expected to change proportionally with the change in the hippocampus TBARS values, but the hippocampus GABA/glutamic acid ratio showed a higher value or higher tendency in the groups with lower TBARS values, such as the groups of P8 + H and P8 + G, than in the P8 group with a higher TBARS value. These findings show that the GABA/glutamic acid ratio could become a marker by which to evaluate aging and/or cognitive impairment in mice. The GABA/glutamic acid ratio in the R1 + H group in comparison with the R1 group also showed a tendency to be higher in the R1 + H group with lower TBARS than the R1 group with higher TBARS. In addition, groups higher in their GABA/glutamic acid ratio showed lower values in their aging scores when compared among to using SAMP8 (see Figure 2 and Figure 4). These results suggest that the GABA/glutamic acid ratio could be used as a marker for the evaluation of aging and/or cognitive impairment in other mice as well as SAMP8.

An increase in the hippocampus GABA content in the P8 + H, P8 + G, and P8 + F groups in the comparison with the P8 group (see Figure 5), though not to a statistically significant level, may indicate that an increase in the level of GABA in the hippocampus may be related to the lowering of TBARS, and that peanuts, their germ-rich fraction, and ferulic acid (ferulic acid is contained as an ester in peanuts) can increase the GABA level.

### 3.3. Regulation of Glial Protein Degradation and Antioxidant Enzymes by Dietary Peanuts

The number of proteins and/or peptides with an accession number, which were identified in differential expression analyses and registered on the NCBI database, was 1400–1600. As there were many proteins identified based on the NCBI database, only proteins that showed significant changes by the differential expression analysis of the protein and/or peptides, which are reported to be influenced in the hippocampus of aged mice and rats, and those of SAMP8 as a model of cognitive impairment [5,6,22,23,24,25] are listed in Table 2. The *p* values, which are higher and lower than 0.95 and 0.05, respectively, show that the protein amounts determined by proteomics are significantly higher and lower in the sample-added group compared to the control, or in the P8 group compared to the R1 group.

Glial fibrillary acidic protein, which is reported to be increased in the brain of mice with accumulated beta amyloid and/or aged SAMP8 [5,26], was significantly higher in the P8 group compared to the R1 group (see P8/R1 in glial fibrillary acidic protein in Table 2) as the control, and the fold change in the P8 + H group to the P8 group decreased from 1.52 to 1.15. The fold change in the P8 + G to P8 group and in the P8 + F to P8 group decreased from 1.52 to 1.16 and from 1.52 to 0.93, respectively, as determined by differential expression analysis. These results showed that the glial acidic protein level in the hippocampus also could become a marker of the peroxidation of hippocampus and aging, as reported by Wu et al. [26]. In addition, the degradation of glial fibrillary protein by the attack of active oxygen species, which might be produced abundantly in SAMP8 compared to SAMR1, may be caused by an increase in the glial fibrillary acidic protein.

As an increase in TBARS in the hippocampus of SAMP8 in comparison with SAMR1 was suppressed by the feeding of peanut wholegrain and its germ-rich fraction, we examined by proteomics what kind of antioxidative enzymes concerned with the scavenging of active oxygen species are regulated by feeding with samples in SAMP8. As it is known that the Mn-SOD level in the brain cortex is higher in SAMP8 mice that are 15 weeks old than in SAMR1 mice that are 8 weeks old [27], the effect of peanut intake on the hippocampal SOD level was determined. The protein expression of the hippocampal SOD tended to be higher in the SAMP8 group fed peanuts and the germ-rich fraction than the SAMP8 groups which were not given these samples, but not to a statistically significant level, indicating that an increase in the expression of SOD by feeding the samples may be related to the lowering of the TBARS values (see P8 + H/P8 and P8 + G/P8 in the SOD in Table 2).

The tendency of the suppression of the protein expression of glutathione S-transferase (GST) in the P8+ H and P8 + G groups, compared to the P8 group (see fold changes of 0.94 and 0.83 in P8 + H/P8 and P8 + G/P8 in Table 2), may indicate that the feeding of the samples may partly suppress aging by regulating GST, because it is known that the hippocampal GST levels generally increase during aging [6].

### 3.4. Regulation of the Expression of Proteins Related to Glutamic Acid Transport and Metabolism by Dietary Peanut

A decreasing tendency in the fold change of protein expression in the solute carrier family 1 (glial high-affinity glutamate transporter; member 2 and 3), and a decreasing tendency in the fold change of the protein expression in the calcium/calmodulin-dependent protein kinase type II (subunit alpha, beta), are both compared to those of the P8 group by feeding peanut and germ-rich fraction, which may indicate that the amount of glutamic acid in the synapse was decreased by feeding peanuts and the germ-rich fraction, resulting in decreases in the amount of glutamic acid which could be transported and in the down-regulation of glial high-affinity glutamate transporter in the glial cell. No change in the fold change of the glutamic synthase by feeding samples is considered to be reflected by the lower glutamic acid concentration in the hippocampus. These results may also indicate that the degree of fold changes in the expression of these enzymes could become a marker for the determination of aging and cognitive impairment.

Calcium/calmodulin-dependent protein kinase type II (CaM kinase II) is known as one of the most prominent protein kinases and is most concentrated in the brain. Neuronal CaM kinase II is widely distributed in the brain and regulates neuronal functions, including the synthesis of neurotransmitters such as glutamic acid, neurotransmitter release, the modulation of ion channel activity, cellular transport, cell morphology, neurite extension, synaptic plasticity, and learning and memory [28], suggesting the possibility that the concentration of glutamic acid in the hippocampus may be related to the expression of CaM kinase II. Based on that information, the effects of peanuts and their germ-rich fraction on the gene expression of CaM kinase II was determined. A significant increase or increase tendency in the protein content of the CaM kinase II in the P8 group compared to the R1 group (fold change in calcium/calmodulin-dependent protein kinase in Table 2, 1.01–1.3) was significantly suppressed in the P8 + H group compared to the P8 group (fold change: 0.7–0.64), but not to a significant level in the P8 + G groups compared to the P8 group (fold change: 0.94–0.89), indicating that peanut has the ability to suppress the expression of CaM kinase II, and its effect may be stronger in peanut itself than in the germ-rich fraction.

The amounts of sodium- and chloride-dependent GABA transporter, which showed a tendency to be decreased in the P8 group in the comparison with R1 group (see the fold change in sodium- and chloride- dependent GABA transporter in Table 2: 0.92), showed a tendency to be increased in the (P8 + H) and (P8 + G) groups compared to the P8 group (see fold change: each, 1.13, 1.31), and a significant increase in the P8 + F compared to the P8 group. These results may indicate that feeding of peanut and germ-rich fraction and ferulic acid could increase the expression of this protein, resulting in an increase-tendency in hippocampal GABA level and GABA/Glu ratio. In addition, the expression of glutamine synthase, which may be concerned with metabolism of glutamic acid in glial cells was not statistically changed by intake of peanut and its germ-rich fraction.

### 3.5. Regulation of the Expression of Protein Related to Formation of Neuromodulator by Dietary Peanut

It is known that tyrosine 3-monooxygenase/tryptophan 5-monooxygenase is a rate-limiting enzyme in the formation of noradrenalin, adrenalin, and serotonin as neuromodulators and that it needs the tyrosine 3-monooxygenase/tryptophan 5-monooxygenase activation protein when it acts, and further that it requires pteridine as a co-factor. Therefore, it is interesting to determine how tyrosine 3-monooxygenase/tryptophan 5-monooxygenase activation protein and dihydropteridine reductase, which is necessary for the formation of pteridine, are influenced by feeding peanut and its germ-rich fraction. A suppression tendency in the fold change in the expression of the 3-monooxygenase/tryptophan 5-monooxygenase activation protein by feeding sample (see 3-monooxygenase/tryptophan 5-monooxygenase activation protein in Table 2: 1.03, 0.82, 0.81; 1.04→0.909), and an increase-tendency in the fold change of dihydropteridine reductase by feeding samples (compared without sample), (see fold change in Table 2: 1.28, 1.55) may indicate that fold change in these proteins in the hippocampus could become a marker for the evaluation of aging and cognitive impairment, and that peanuts and its germ-rich fraction may be able to reduce dysfunction in brain.

As it is reported that synapsin-2 is concerned with the formation and maintenance of synapses in the culture of hippocampus [29] and with age-dependent cognitive impairment [29,30], the effect of peanut intake on the expression of hippocampal synapsin-2 level was determined. Although little difference in the synapsin 2 level was observed between the R1 and P8 groups, the fold changes of P8 + H to P8 and P8 + G to P8 in synapsin-2 level showed a significantly lower level (see synapsin-2 in Table 2: 0.77, 0.79 with *p* value of 0), suggesting that peanuts and their germ-rich fraction could inhibit increase in the synapsin-2 level, which might be involved with aging. This result also suggests that the hippocampus synapsin-2 level could also become a marker for the evaluation of aging and cognitive impairment and for the development of food available to protect against aging and cognitive impairment, as suggested by Asamoto et al.

### 3.6. Tublin and Hemoglobin Subunit Levels in Hippocampus of Mice Given Peanut

Hippocampus tublin-alpha is considered to be important for the preservation of plasticity of the brain, and it is reported that its levels in the hippocampus are lower in old rats than younger rats [5]. A value of less than one in the fold change of tublin alpha in the P8 + H/P8 and P8 + G/P8 groups (see tublin alpha in Table 2: each, 0.96 (*p*: 0), 0.97 (*p*: 0.04) value), may indicate that the feeding of peanut and its germ-rich fraction may not able to improve the plasticity of the brain.

### 3.7. Hemoglobin Subunit Alpha and Beta Levels in Hippocampus of Mice Given Peanuts

As it is known that the protein expression of hemoglobin subunit alpha and beta are lower in the hippocampus of old rats than young ones when measured by proteomics [5], the dietary intake of peanuts and their germ-rich fraction on their hippocampus levels were determined. As the amount of hemoglobin alpha and beta in the P8 group was lower than that of the R1 group when evaluated from the fold change in the P8/R1 (see fold change in the R1/P8 in hemoglobin subunit in Table 2: 0.79–0.8), and as the fold change in the (P8 + H)/P8 and (P8 + G)/P8 increased (each, 1.2–1.12, 1.19–1.11) when the peanut and germ-rich fraction were given, it was indicated that hemoglobin alpha and beta are protected from their decrease by the feeding of peanut and its germ-rich fraction to SAMP8. This result also indicated that the expression of these proteins in the hippocampus also can be used as a marker for evaluating progress in aging and cognitive impairment in SAMP8.

### 3.8. Discussion

Although, to date, a few reports have dealt with the possibility that peanuts may improve cognitive disorders in middle-aged and elderly subjects [31], there are no reports which have shown clearly protective effects of peanuts against aging and cognitive disorders in humans as well as SAMP8. Some of these results were caused by the weak effect of peanuts and/or low number of markers used for the evaluation of the protective effects in those reports, suggesting the necessity of many markers which could indicate the effects of peanuts with greater ease and clarity. The regulation of many markers (glial high-affinity glutamate transporter, calcium/calmodulin-dependent protein kinase type II, sodium- and chloride-dependent GABA transporter, glial fibrillary acidic protein, synapsin-2, tyrosine 3-monooxygenase/tryptophan 5-monooxygenase activation protein and dihydropteridine reductase) which could be examined by proteomics, and hippocampus (GABA/glutamic acid) ratio which increased in SAMR1 and SAMP8 fed with peanuts, may confirm the effect of peanuts in protecting aging and cognitive impairment, but the effects of peanuts for improvement in appearance such as glossiness, coarseness, loss of hair, skin ulcers, and lordokyphosis of the spine could not observed clearly. To confirm the protective effects of the high-oleic acid against cognitive impairment, it may be necessary to examine its effects on parameters related to aging and cognitive impairment in organs and tissues other than the hippocampus. The effects of the high-oleic peanuts on protein expression in the liver will be examined in future by using proteomics. The feeding of peanuts for longer periods also might be necessary to find the effects on the appearance of the back coat of SAMP8.

As SAMR1 and SAMP8 fed with high-oleic peanuts showed lower hippocampus TBARS levels, a delay in oxidation in vivo and/or in the hippocampus in mice fed with high-oleic peanuts may be related to the effect of peanuts. A low content of fatty acid, which is prone to oxidation, and the presence of anti-oxidative polyphenols such as p-coumaloyl tartrate and feruloyl tartrate in high-oleic peanuts are also considered to have contributed to the suppression of oxidation. However, more experiments may be necessary to clarify the action mechanism of high-oleic peanuts, including the mechanism of its effect on the protein expression of markers.

## 4. Conclusions

As summarized in Table 3 the present study suggested that changes in the aging score, TBARS, glial fibrillary acidic protein, GABA/glutamic acid ratio, glial high-affinity glutamate transporter, GST, calcium/calmodulin-dependent protein kinase type II, 3-monooxygenase/tryptophan 5-monooxygenase activation protein, dihydropteridine reductase, synapsin-2, tublin-alpha, and hemoglobin subunit beta and alpha in the hippocampus of SAMP8 could become markers which can evaluate the progress of aging and cognitive impairment. In particular, the GABA/glutamic acid ratio could become a new and sensitive marker, which is considered to be a remarkable finding in the present study. Peanuts (wholegrains) and their germ-rich fraction were considered as capable of preventing aging and cognitive impairment when evaluated by many of the above markers. Ferulic acid, which is contained as an ester and minor component in peanut, could be partly related to the effect of peanut. Although the germ-rich fraction showed a tendency to be slightly superior to peanut wholegrain in the improvement of some of the markers, a clear difference was not observed between the peanut-fed and germ-rich fraction-fed mice. For the clarification of the action mechanisms, studies using the principal compounds which can be isolated from peanuts may be necessary.

## Figures and Tables

**Figure 1 nutrients-12-03461-f001:**
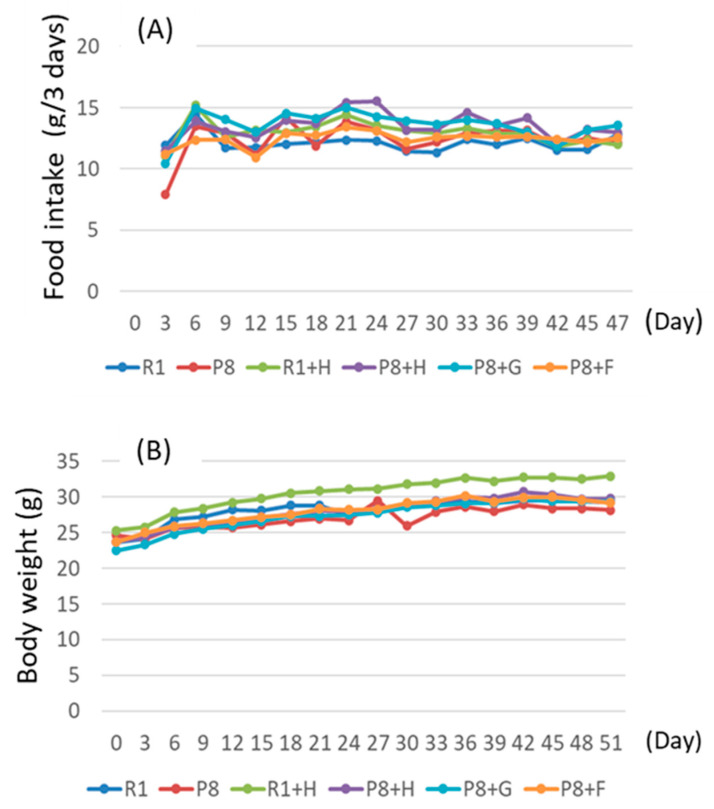
Food intake every 3 days (**A**) and changes in body weight gain (**B**) in mice given peanut wholegrains, germ-rich fractions, and ferulic acid. R1: senescence-accelerated resistant mice (SAM-R1) group fed with basal diet; P8: senescence-accelerated mouse prone/8 (SAMP8) group fed with basal diet; R1 + H: SAM-R1 group fed with basal diet with added peanut wholegrains; P8 + H: SAMP8 group fed with basal diet with added peanut wholegrains; P8 + G: SAMP8 group fed with basal diet with added germ-rich fraction of wholegrain; P8 + F: SAMP8 group fed with basal diet with added ferulic acid. Food intake every 3 days and body weight are the means of each group.

**Figure 2 nutrients-12-03461-f002:**
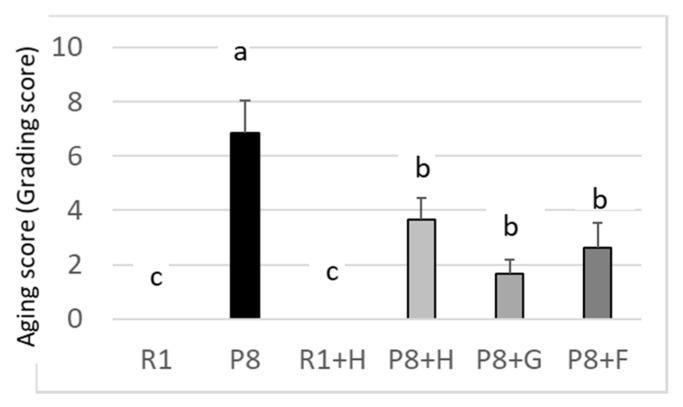
Effects of dietary peanut, germ-rich fraction, and ferulic acid on the aging score of SAMR1 and SAMP8. Aging score was tentatively calculated from the degree of disappearance of the glossiness of the back coat and the degree of lordokyphosis of the spine appearance. Five grades (1–5: from weaker to stronger) were used depending on the severity of appearance. R1: senescence-accelerated resistant mice (SAM-R1) group fed with basal diet; P8: senescence-accelerated mouse prone/8 (SAMP8) group fed with basal diet; R1 + H: SAM-R1 group fed with basal diet with added peanut wholegrain; P8 + H: SAMP8 group fed with basal diet with added peanut wholegrains; P8 + G: SAMP8 group fed with basal diet with added germ-rich fraction of wholegrain; P8 + F: SAMP8 group fed with basal diet with added ferulic acid. Groups without a common letter differ significantly (*p* < 0.05).

**Figure 3 nutrients-12-03461-f003:**
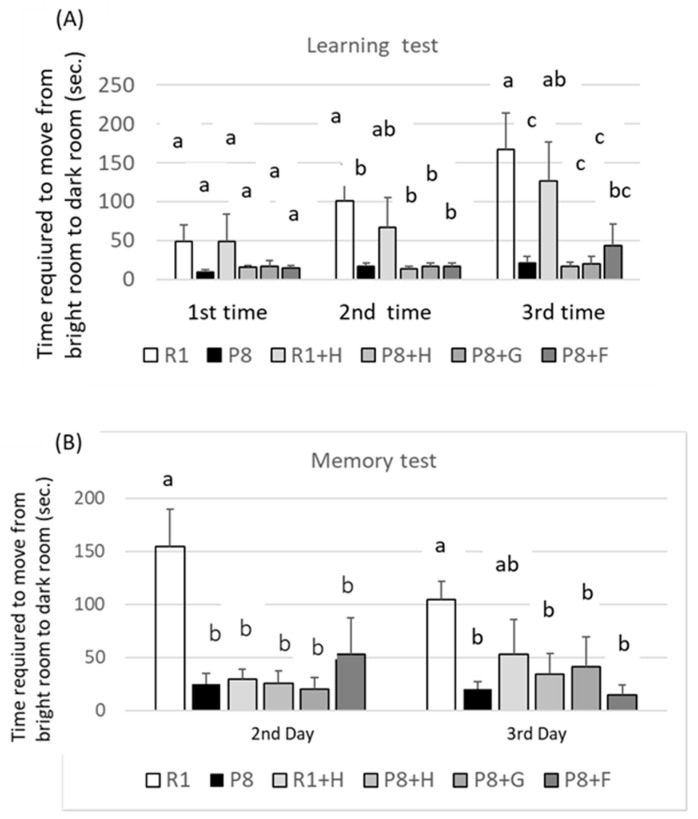
Effects of dietary peanut, its germ-rich fraction, and ferulic acid on the learning (**A**) and memory (**B**) abilities of SAMR1 and SAMP8. sec. in vertical line means seconds. R1: senescence-accelerated resistant mice (SAM-R1) group fed with basal diet; P8: senescence-accelerated mouse prone/8 (SAMP8) group fed with basal diet; R1 + H: SAM-R1 group fed with basal diet with added peanut wholegrain; P8 + H: SAMP8 group fed with basal diet with added peanut wholegrains; P8 + G: SAMP8 group fed with basal diet with added germ-rich fraction of wholegrain; P8 + F: SAMP8 group fed with basal diet with added ferulic acid. Groups without a common letter differ significantly (*p* < 0.05).

**Figure 4 nutrients-12-03461-f004:**
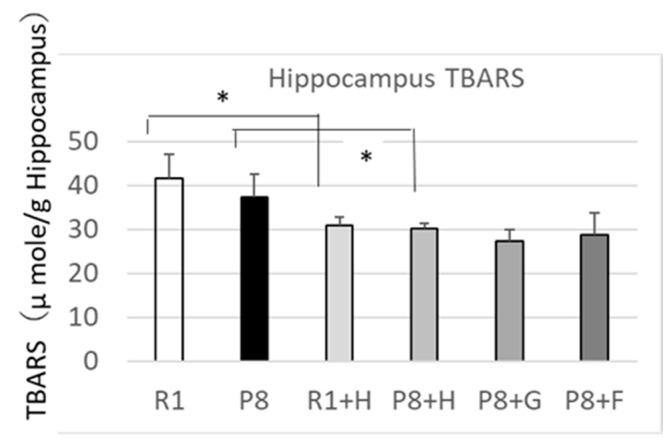
Effect of dietary peanut, germ-rich fraction, and ferulic acid on the hippocampus TBARS in SAMP8. R1: senescence-accelerated resistant mice (SAM-R1) group fed with basal diet; P8: senescence-accelerated mouse prone/8 (SAMP8) group fed with basal diet; R1 + H: SAM-R1 group fed with basal diet with added peanut wholegrain; P8 + H: SAMP8 group fed with basal diet with added peanut wholegrains; P8 + G: SAMP8 group fed with basal diet with added germ-rich fraction of peanut wholegrain; P8 + F: SAMP8 group fed with basal diet with added ferulic acid. * differences between the two groups were significant when compared by the Student’s *t*-test (*p* < 0.05). TBARS: Thiobarbituric acid reactive substances.

**Figure 5 nutrients-12-03461-f005:**
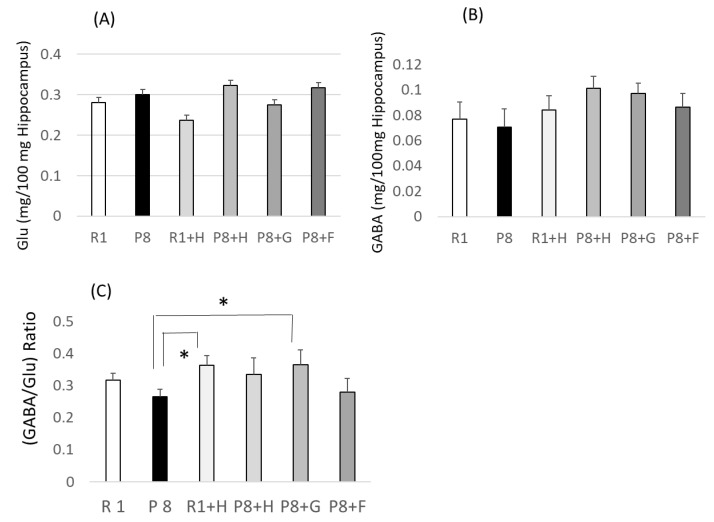
Effect of dietary peanuts and ferulic acid on hippocampus glutamic acid (**A**) and GABA (**B**) contents, and on (GABA/Glutamic acd) ratio (**C**) in SAMR1 and SAMP8. R1: senescence-accelerated resistant mice (SAM-R1) group fed with basal diet; P8: senescence-accelerated mouse prone/8 (SAMP8) group fed with basal diet; R1 + H: SAM-R1 group fed with basal diet with added peanut wholegrain; P8 + H: SAMP8 group fed with basal diet with added peanut wholegrains; P8 + G: SAMP8 group fed with basal diet with added germ-rich fraction of wholegrain; P8 + F: SAMP8 group fed with basal diet with added ferulic acid. * significantly different when compared by Student’s *t*-test (*p* < 0.05).

**Table 1 nutrients-12-03461-t001:** Composition of diet (%).

Expreimental Groups	R1	P8	R1 + H	P8 + H	P8 + G	P8 + F
Mouse strains	SAMR1	SAMP8	SAMR1	SAMP8	SAMP8	SAMP8
Casein	20	20	20	20	20	20
α-Cornstarch	55.95	55.95	52.47	52.47	52.03	55.75
Sucrose	7	7	7	7	7	7
Cellulose powder	5	5	5	5	5	5
Corn oil	7	7	3.48	3.48	3.92	7
Mineral mixture ^a^	3.5	3.5	3.5	3.5	3.5	3.5
Vitamin mixture ^b^	1	1	1	1	1	1
L-Cystine	0.3	0.3	0.3	0.3	0.3	0.3
Choline bitartrate	0.25	0.25	0.25	0.25	0.25	0.25
Peanuts wholegrains			7	7		
Germ-rich fraction					7	
Ferulic acid						0.2
Total (%)	100	100	100	100	100	100

R1: senescence-accelerated resistant mice (SAM-R1) group fed with basal diet; P8: senescence-accelerated mouse prone/8 (SAMP8) group fed with basal diet; R1 + H: SAM-R1 group fed with basal diet with added peanut wholegrain; P8 + H: SAMP8 group fed with basal diet with added peanut wholegrains; P8 + G: SAMP8 group fed with basal diet with added germ-rich fraction of wholegrain; P8 + F: SAMP8 group fed with basal diet with added ferulic acid. Both ^a^ AIN-93G-MX and ^b^ AIN-93-VX a were purchased from Clea Japn (Tokyo, Japan).

**Table 2 nutrients-12-03461-t002:** Effects of the feeding of peanut wholegrain, its germ-rich fraction, and ferulic acid on the amounts of protein expression in the hippocampus of SAMP8.

Accession	Description		(R1 + H)/R1	P8/R1	(P8 + H)/P8	(P8 + G)/P8	(P8 + F)/P8
	**Glial fibrillary acidic protein**						
GFAP_MOUSE	Glial fibrillary acidic protein OS = Mus musculus GN = Gfap PE = 1 SV = 4						
		Fold change (ratio)	1.2	1.52	1.15	1.16	0.93
		*p* Value	0.94	1	0.98	0.99	0.16
A0MTM0_MOUSE	Glial fibrillary acidic protein (Fragment) OS = Mus musculus GN = Gfap PE = 2 SV = 1						
		Fold change (ratio)			1.28	1.09	1.07
		*p* Value			0.98	0.72	0.69
	**SOD**						
Q3U8W4_MOUSE	Superoxide dismutase OS=Mus musculus GN = Sod2 PE = 2 SV = 1						
		Fold change (ratio)	NU	NU	1.16	1.12	1.04
		*p* Value			0.65	0.65	0.56
Q4FJX9_MOUSE	Superoxide dismutase OS = Mus musculus GN = Sod2 PE = 2 SV = 1						
		Fold change (ratio)	NU	NU	1.25	1.12	1.12
		*p* Value			0.74	0.62	0.64
SODM_MOUSE	Superoxide dismutase [Mn], mitochondrial OS = Mus musculus GN = Sod2 PE = 1 SV = 3						
		Fold change (ratio)	NU	NU	1.23	1	1.16
		*p* Value			0.73	0.5	0.62
	**Glutathione S-transferase**						
D3YVP6_MOUSE	Glutathione S-transferase Mu 7 OS = Mus musculus GN = Gstm7 PE = 4 SV = 2						
		Fold change (ratio)	1.02	1.04	0.94	0.83	0.89
		*p* Value	0.59	0.67	0.33	0.12	0.2
D3YX76_MOUSE	Glutathione S-transferase Mu 2 OS = Mus musculus GN = Gstm2 PE = 3 SV = 1						
		Fold change (ratio)	1.03	1.02	1.04	0.83	
		*p* Value	0.57	0.63	0.6	0.08	
A2AE91_MOUSE	Glutathione S-transferase, mu 4 OS = Mus musculus GN = Gstm4 PE = 3 SV = 1						
		Fold change (ratio)	1.01	1.05	0.93	0.83	0.85
		*p* Value	0.52	0.69	0.3	0.11	0.1
Q8R5I6_MOUSE	Glutathione S-transferase mu 4 OS = Mus musculus GN = Gstm4 PE = 2 SV = 1						
		Fold change (ratio)	1.02	1.09	0.92	0.81	0.91
		*p* Value	0.55	0.83	0.3	0.08	0.25
	**Glial high affinity glutamate transporter**						
A2APL7_MOUSE	Solute carrier Family 1 (Glial high affinity glutamate transporter), member 2 OS = Mus musculus GN = Slc1a2 PE = 2 SV = 1						
		Fold change (ratio)	0.72	0.91	1.04	0.86	0.78
		*p* Value	0.02	0.22	0.61	0.14	0.16
A2APL8_MOUSE	Solute carrier family 1 (Glial high affinity glutamate transporter), member 2 OS = Mus musculus GN = Slc1a2 PE = 4 SV = 1						
		Fold change (ratio)	0.73	1.05	0.93	0.73	1.25
		*p* Value	0.01	0.61	0.29	0.02	0.92
Q543U3_MOUSE	Solute carrier family 1 (Glial high affinity glutamate transporter), member 3 OS = Mus musculus GN = Slc1a3 PE = 2 SV = 1						
		Fold change (ratio)	0.93	1.19	0.82	0.84	1.07
		*p* Value	0.38	0.87	0.13	0.17	0.63
A2APM7_MOUSE	Solute carrier family 1 (Glial high affinity glutamate transporter), member 2 OS = Mus musculus GN = Slc1a2 PE = 4 SV = 1						
		Fold change (ratio)	0.79	0.94	0.63	0.41	1.45
		*p* Value	0.11	0.31	0.15	0	0.96
Q3UYK6_MOUSE	Solute carrier family 1 (Glial high affinity glutamate transporter), member 2 OS = Mus musculus GN = Slc1a2 PE = 2 SV = 1						
		Fold change (ratio)	0.79	0.81	0.94	0.76	1.25
		*p* Value	0.08	0.19	0.35	0.06	0.94
	**Calcium/calmodulin-dependent protein kinase**						
KCC2B_MOUSE	Calcium/calmodulin-dependent protein kinase type II sub-unit beta OS = Mus musculus GN = Camk2b PE = 1 SV = 2						
		Fold change (ratio)	0.99	1.3	0.64	0.94	1.17
		*p* Value	0.39	0.99	0	0.27	0.99
KCC2A_MOUSE	Calcium/calmodulin-dependent protein kinase type II sub-unit alpha OS = Mus musculus GN = Camk2a PE = 1 SV = 2						
		Fold change (ratio)	0.97	1.01	0.85	0.89	1.11
		*p* Value	0.3	0.5	0.03	0.06	0.98
E9Q1V9_MOUSE	Calcium/calmodulin-dependent protein kinase type II sub-unit delta OS = Mus musculus GN = Camk2d PE = 4 SV = 1						
		Fold change (ratio)	0.93	1.07	0.7	0.91	1.2
		*p* Value	0.21	0.86	0	0.22	0.99
	**Sodium- and chloride-dependent GABA transporter**						
S6A11_MOUSE	Sodium- and chloride-dependent GABA transporter 3 OS = Mus musculus GN = Slc6a11 PE = 1 SV = 2						
		Fold change (ratio)	1.14	0.92	1.13	1.31	1.67
		*p* Value	0.78	0.29	0.86	0.9	0.99
	**Glutamine synthase**						
GLNA_MOUSE	Glutamine synthetase OS = Mus musculus GN = Glul PE = 1 SV = 6						
		Fold change (ratio)	0.9	1	1.01	0.91	0.83
		*p* Value	0.23	0.51	0.56	0.27	0.06
	**Tyrosine 3-monooxygenase/tryptophan 5-monooxygenase activation protein**						
A2A5N2_MOUSE	Tyrosine 3-monooxygenase/tryptophan 5-monooxygenase activation protein,						
	beta OS = Mus musculus GN = Ywhab PE = 2 SV = 1						
		Fold change (ratio)	0.95	1.03	0.82	0.81	0.88
		*p* Value	0.24	0.75	0	0.01	0.05
Q5SS40_MOUSE	Tyrosine 3-monooxygenase/tryptophan 5-monooxygenase activation protein,						
	epsilon polypeptide OS = Mus musculus GN = Ywhae PE = 2 SV = 1						
		Fold change (ratio)	1.01	1.04	0.9	0.9	0.89
		*p* Value	0.53	0.83	0.04	0.03	0.01
	**Dihydropteridine reductase**						
D3YWR7_MOUSE	Dihydropteridine reductase OS = Mus musculus GN = Qdpr PE = 4 SV = 1						
		Fold change (ratio)	NU	NU	NU	1.28	NU
		*p* Value				0.79	
DHPR_MOUSE	Dihydropteridine reductase OS = Mus musculus GN = Qdpr PE = 1 SV = 2						
		Fold change (ratio)	NU	NU	NU	1.55	NU
		*p* Value				0.88	
	**Tubulin alpha**						
TBA1A_MOUSE	Tubulin alpha-1A chain OS = Mus musculus GN = Tuba1a PE = 1 SV = 1						
		Fold change (ratio)	0.99	1.08	0.96	0.97	0.96
		*p* Value	0.19	1	0	0.04	0.03
TBA1C_MOUSE	Tubulin alpha-1C chain OS = Mus musculus GN = Tuba1c PE = 1 SV = 1						
		Fold change (ratio)	0.99	1.07	0.95	0.97	0.97
		*p* Value	0.39	1	0.01	0.02	0.05
O89052_MOUSE	Alpha-tubulin (Fragment) OS = Mus musculus GN = Tuba1b PE = 2 SV = 1						
		Fold change (ratio)	1	1.16	0.9	0.97	1.02
		*p* Value	0.46	1	0	0.12	0.77
	**Synapsin-2**						
SYN2_MOUSE	Synapsin-2 OS = Mus musculus GN = Syn2 PE = 1 SV = 2						
		Fold change (ratio)	0.84	0.97	0.77	0.79	1.01
		*p* Value	0.01	0.31	0	0	0.59
	**Hemoglobin subunit alpha**						
HBA_MOUSE	Hemoglobin subunit alpha OS = Mus musculus GN = Hba PE = 1 SV = 2						
		Fold change (ratio)	0.97	0.8	1.2	1.19	1.06
		*p* Value	0.31	0	1	1	0.88
	**Sodium- and chloride-dependent GABA transporter**						
S6A11_MOUSE	Sodium- and chloride-dependent GABA transporter 3 OS = Mus musculus GN = Slc6a11 PE = 1 SV = 2						
		Fold change (ratio)	1.14	0.92	1.13	1.31	1.67
		*p* Value	0.78	0.29	0.86	0.9	0.99
	**Hemoglobin subunit beta**						
HBB1_MOUSE	Hemoglobin subunit beta-1 OS = Mus musculus GN = Hbb-b1 PE = 1 SV = 2						
		Fold change (ratio)	0.94	0.79	1.14	1.17	0.99
		*p* Value	0.06	0	1	1	0.37
E9Q223_MOUSE	Hemoglobin subunit beta-1 (Fragment) OS = Mus musculus GN = Hbb-b1 PE = 3 SV = 1						
		Fold change (ratio)	0.9	0.79	1.12	1.11	0.92
		*p* Value	0.01	0	0.97	0.97	0.1
HBB2_MOUSE	Hemoglobin subunit beta-2 OS = Mus musculus GN = Hbb-b2 PE = 1 SV = 2						
		Fold change (ratio)	0.98	0.84	1.12	1.16	0.99
		*p* Value	0.32	0	0.98	1	0.44

R1: senescence-accelerated resistant mice (SAM-R1) group fed with basal diet; P8: senescence-accelerated mouse prone/8 (SAMP8) group fed with basal diet; R1 + H: SAM-R1 group fed with basal diet with added peanut wholegrain; P8 + H: SAMP8 group fed with basal diet with added peanut wholegrains; P8 + G: SAMP8 group fed with basal diet with added germ-rich fraction of wholegrain; P8 + F: SAMP8 group fed with basal diet with added ferulic acid. Accession or number is a sries of codes or numbers that are assigned to the identified protein by NCBI. Fold change is expressed as the ratio between two groups (NU group/DE group). *p* values higher than 0.95 show that the protein content is statistically higher in the numerator group than the denominator group. *p* values lower than 0.05 show that protein content is statistically higher in the denominator group than that of the numerator group. Columns filled with NU and DE show that proteins (enzymes) corresponding to the column were detected in only the NU and DE groups, respectively. PE, protein evidence; SV, structural variance; GABA, gannma-aminobutylic acid, SOD, superoxide dismutase.

**Table 3 nutrients-12-03461-t003:** Amino acid ratio and proteins could become markers for the determination of aging and cognitive impairment in SAMP8 and its regulation by intake of high-oleic peanuts (peanuts and their germ-rich fraction).

Amino Acid Ratio and Proteins Could Become Markers Used for Evaluation of Aging Status and Cognitive Impairment	Expression	
	Expression in SAMP8, Compared to SAMR1	Expression in SAMP8 Given High Oleic Peanut, Compared to SAMP8
Hippocampus (GABA/glutamic acid) ratio	↓	↑
Glial fibrillary acidic protein	↑	↓
SOD	→	↑
Glutathione S-trnaserase	→	↓
Solute carrier family 1 (Glial high affinity glutamate transporter), member 2	→	↓
Calcium/calmodulin-dependent protein kinase type II sub-unit beta	↑	↓
Calcium/calmodulin-dependent protein kinase type II sub-unit alpha	→	↓
Sodium- and chloride-dependent GABA transporter	↓	↑
Tyrosine 3-monooxygenase/tryptpphan 5-monooxygenase activation protein, beta	↑→	↓
Dihydropteridine reductase	↓→	↑
Tubulin alpha	↑	↓
Synapsin-2	→	↓
Hemoglobin subunit alpha	↓	↑
Hemoglobin subunit beta	↓	↑

The arrows, ↓, ↑, and → show down and up regulations and no-altered state, respectively. GABA, gamma-aminobutylic acid; SOD, superoxide dismutase.

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
