# Peer review of "Effect of High-Oleic Peanut Intake on Aging and Its Hippocampal Markers in Senescence-Accelerated Mice (SAMP8)"

_nutrients, 2020, doi:10.3390/nu12113461_

Round 1
Reviewer 1 Report
Igarashi and Kurata demonstrated the effect of high oleic peanuts and its germ-rich fraction against age-related cognitive impairment in SAMP8 mice. The behavior and several aging markers were detected to evaluate the effect. Finally, they suggested that high oleic peanuts and its germ-rich fraction can protect aging and cognitive impairment via influencing the markers measured in the study.
This is an interesting study. However, the results were not strong enough to support what they claim.
The following are the comments
Major
1. The authors must provide IACUC number of the study.
2. The results of behavior experiment were not strong enough to make a conclusion that high oleic peanuts attenuated the age-related cognitive impairment, because there was no significant different between P8 group and the other SAMP8 mice groups. Therefore, other experiment for assessing cognition must be perform to confirm the effect of oleic peanuts on age-related cognitive decline.
3. The aging markers that were suggested by the authors showed inconsistent trend in the study. For example, R1 group had better performance in learning and memory test than the other SAMP8 mice groups, but the most markers were not higher. Moreover, P8+F seemed to have better performance in behavior tests that the other SAMP8 groups. However, the some markers of P8+F were better but some were not. Thus, more age-related markers must be done to make a more convincing conclusion.
Minor
1. The color of each group in bar chart should be consistent.
2. The authors should discuss the possible reasons why oleic peanuts have the effects on the markers.
Author Response
To Reviewer A
I really thank for helpful many suggestions.
In order to strong the effect of high oleic peanuts and its germ-rich fraction, it might be necessary to carry out the experiment over a longer day. I discussed this point in the [Results and Discussion] of a revised eddition..
Line 378-379. highlighted in yellow.
〇About major points
- Line 83. ・・・・Yamagata University was 2905 →・・・・Yamagata University was 2905 (IACUC number:2905) ・・・ Corrected in revised manuscript.
- Line 159-170. ・・・Electric stimulation to mice SAMP8 might be too weak to detect the differences in the responsiveness of mice ( Line 166-167 in original manuscript). → ・・・As there were helpful suggestions, corrected in revised manuscript as follows.
: For reasons, week effects of sample and lowering in responsibility of SAMP8, compared to SAMR1, for electric stimulation by aging, may be considered. Experiments under more strong electric stimulation might be necessary for confirmation of the effect of high oleic peanuts. ・・・Line 168-170. Highlighted in yellow.
〇About Minor points
- Fig1 was remained without correction, because each line could not discriminated each other when the same strain color was used. Other figures were corrected as suggested.
- As there were comment to discuss the possible action mechanism, new discussion was added to the revised text as bellow. ・・・Line 380-386. Highlighted in yellow.
: As SAMR1 and SAMP8 fed with high oleic peanuts showed lower hippocampus TBARS levels, delay in oxidation in vivo-and/or hippocampus in mice fed with high oleic peanuts may be concerned with the effect of peanuts. Low content of fatty acid which prone to oxidation, and the presence of anti-oxidative polyphenols such as p-coumaloyl tartrate and feruloyl tartrate in the high oleic peanuts is also considered to have contributed to the suppression of oxidation. However, more experiment may be necessary to clarify the action mechanism of high oleic peanuts, including the mechanism for its effect to the protein expression of markers.
3. As there were many suggestions, discussion throughout was carried out in revised text.
Line 365-386. Highlighted in yellow and green.
Best regards
Kiharu igarashi

Reviewer 2 Report
Report on manuscript
Effect of high oleic peanut intake on the aging and its hippocampal markers in senescence-accelerated mice (SAMP8)
- Manuscript ID: 981493
- Journal: Nutrients
The present manuscript by Igarashi and Kurata reports the protective effects of high oleic peanuts and its germ-rich fraction against aging and cognitive impairment in SAMP8 mice. Through proteomic analysis, authors have suggested several markers of senescence and cognitive impairment. From nutritious point of view, study is interesting. However, there are many open questions.
Major Concerns
- Overall, manuscript is not well-written from English Language point of view. Manuscript should be revised with the help of native speaker. In most of the manuscript, authors did not seem to be able to make clear statements & sentences.
- Abstract is not well-structured. It should be re-written.
- There is a weak connection between literature survey and study design in introduction.
- Results are described in a very vague way. Authors were not able to elaborate the results in a clear way. Therefore, it is recommended that results section should be improved more carefully and with more clear statements & claims.
- To evaluate the degree of senescence, study of only the difference in appearance (glossiness and other) seems not enough to make a solid claim about senescence.
- In results section, 3.3. it is stated “Regulation of protein degradation by dietary peanuts”. Which specific protein is degraded (is it glial fibrillary acidic protein?) and how it is degraded? Is degradation of the protein has been analyzed specifically?
Minor concerns
- In materials and methods section, “Sample preparation”, lipid contents were measured. Did authors determine the Oleic acid contents specifically?
- Why Learning and Memory Ability Test was performed only at 46-47 day, nor before and after that? Any explanation for this time selection?
- Table 2, it is protein expression not expretion.
- Reference 5. There should be a full stop after title of the article, not comma.
- Reference 6. Page numbers are missing.
- Reference 5, 6, 11, 12, 13, 17, 26, 28 & 31. Journal abbreviation is wrong.
- Reference 31 and 32 are same.
Author Response
To Reviewer B
I really thank for helpful suggestions.
I will soon ask MDPI”s English editing service to correct English after correction of the points suggested by Reviewers.
I would like to respond to the helpful suggestions as follows.
- English
After correcting all of the points suggested by the referee, we will correct the English by asking MDPI"s English editing service.
- Abstract
The structure of the abstract re-examined and corrected to the extent that I can. I'm going to ask PI"s English editing service for a more appropriate fix.
- A week connection between literature survey and this study
A little in the literature which described clearly the effects of peanuts is one of the reasons why we cannot introduce many relationships between the experimental plan and the literature of this research. The relationship between the literature and the research plan was examined as much as possible and introduced in introduction of the original manuscript. ・・・Line 40-53
- Reason why could not describe the results in clear way.
As there were helpful suggestion in discussion of the results, below sentence was added to last of [Results and Discussion] in revised manuscript. Highlighted in green (Line 361-378 in revised manuscript). As there was suggestion by reviewer A, next yellow part (Line 378-386) was newly prepared and added. ・・・Line 378-386. Highlighted in green.
Although, till now, there is a few reports dealt with the possibility that peanuts may have improvement effect for cognitive disorders in middle-aged and elderly subjects [32], there is no reports which showed clearly protective effects of peanuts against aging and cognitive disorders in human as well as SAMP8. Some of these results were caused by weak effect of peanuts and/or low in number of markers used for evaluation of the protective effects in those reports, suggesting necessity many markers which could find the effects of peanuts more easily and clearly. Regulation of many markers (Glial high affinity glutamate transporter, calcium/calmodulin-dependent protein kinase type II, sodium- and chloride-dependent GABA transporter, glial fibrillary acidic protein, synapsin-2, tyrosine 3-monooxygenase/tryptophan 5-monooxygenase activation protein and dihydropteridine reductase) which could examined by proteomics, and hippocampus (GABA/glutamic acid) ratio by dietary peanuts, may confirm the effect of peanuts in protecting aging and cognitive impairment, but the effects of peanuts for improvement in the appearance such as glossiness, coarseness, loss of hair, skin ulcers, and lord kyphosis of the spine could not observed clearly. Feeding of peanuts for longer periods might be necessary to find the effects on appearance in back coat of SAMP8.
〇Reference [31] in revised manuscript was added newly. ・・・Highlighted in yellow.
[31] Rusu, M.e.; Mocan, A.; Ferreira, I.C.F.R.; Popa, D.S. Health benefits of nut consumption in middle-agged and elderly population. Antioxidants 2019, 8, 302. Doi: 10.3390/antiox8080302.
- Reglulation of protein degradation
Added a missing word (glial). Line 240. Highlighted in green.
As there was helpful suggestion, next sentence, which may help in reducing the mechanism for an increase in glial fibrillary acidic protein, was added to the revised manuscript as bellow. ・・・ Line 256-258 in revised manuscript. Highlighted in green.
In addition, degradation of glial fibrillary protein by attack of active oxygen species which might be produced abundantly in SAMP8, compared to SAMR1, may be cause in an increase in glial fibrillary acidic protein.
- Analysis of glial fibrillary acidic protein.
Glial fibrillary acidic protein was not analyzed by chemical analysis in this study, but we can reduce the changes in the protein expression from variation analysis in proteomics.
Minor concerns
- Oleic acid
The amount of oleic acid of peanuts used in this experiment was not-measured in this experiment. However, before exporting to Japan, oleic acid content was determined in USA and its content was confirmed (over 70%). For this reason, only the oleic acid content in the peanuts used in this experiment was written in the text. (in parentheses). ・・・ Line 69 in revised manuscript. Highlighted in green.
- Selection of date for learning and memory ability test
Next sentence was added to help understanding of date set for memory and learning test.
: Learning test and memory availability test was set to finish by one week of the dissection date to avoid stress on the mouse as much as possible, and not to affect various measurements data after dissection. ・・・ Line 100-102. Highlighted in green.
- Protein expression in Table 2
Spelling error was corrected. ・・・ Line 270. Highlighted in green.
- Reference 5.
A full stop after title was corrected.
. Ageing Res. Rev. 2009, 8, 1-17. ・・・Line 426. Highlighted in green.
- Reference Missing of page numbers
- Neurochem. 1980, 35, 1246-1249.・・・ Line 428. Highlighted in green.
- Reference 11.
Brain Res. 2009, 256, 111-122. ・・・ Line 443. Highlighted in green.
- Reference 12
Exp. Gerontol. 2005,40,774-783 ・・・ Line 446. Highlighted in green.
- Reference 13
Behav. Brain Res.2017,322,288-298. ・・・ Line 449. Highlighted in green.
- Reference 17
Biol. Pharm. Bull.2010,33,759-767. ・・・ Line 459. Highlighted in green.
- Reference 26
No.26 in original manuscript was a duplicate of No.5. So. It was removed in the revised manuscript.
- Reference 27
Neurosci. Lett. 2001, 298, 135-138.・・・ Line 485. Highlighted in green.
- Reference 30
J.Cell.Sci.2008, ・・・ Line 492. Highlighted in green.
- Reference 31
Antioxidants 2019, 8, 302. Doi: 10.3390/antiox8080302. ・・・Line 494. Newly added. Highlighted in yellow.
Best regards
Kiharu Igarashi

Round 2
Reviewer 1 Report
The revised manuscript was significantly improved. However, the key concern of mine is still there. Although the authors measured several biomarkers in hippocampus to show the effects of high oleic peanut, there was no difference between the P8 control group and other P8 groups in behavior tests. The only difference of phenomenon was the aging score that contained many parameters not related to age-related cognitive impairment. Therefore, how the high oleic peanut attenuated age-related declines other than brain function is necessary to clarify the real effect of high oleic peanut on aging.
Author Response
I really thank for your helpful discussion and suggestions.
We carried out the experiment before this experiment, which was almost the same with this experiment, but feeding period was 45 days. As we could not observed clear difference in phenomenon in that experiment, we planed this experiment and prepared this report. In the previous experiment (45 days experiment), it was observed that protein expression of several proteins in the liver, which was determined by proteomics and difference analysis of SAMR1 and SAMP8, was regulated by feeding high oleic acid peanuts (especially, major urinary protein, catalase, ferritin), suggesting that the other parameters other than hippocampus may be useful for confirmation of the effect of high oleic peanuts. We will start analysis of liver and kidneys by using proteomics and the other methods. Thanks for very helpful suggestions.
For those reason, the revised version added a new sentence to discussion as shown in yellow color in the Revised manuscript (Edited by MDPI).
Best regards
Kiharu Igarashi

Reviewer 2 Report
Manuscript has been improved than first revision but still there is space to improve it to make it more presentative and convincing for the audience. I will also suggest to take help of English Editing Service by any native speaker and then I should work well.
Author Response
I really thank for your helpful discussion and suggestions in improvements of our manuscript.
Reviewer 1 pointed out that it was useful, so in the revised version, I added one new sentence as shown in the revised manuscript (Edited by MDPT, highlighted in yellow in the revised manuscript)
Best regards
Kiharu Igarashi
